

# Sequential lexicon enhanced bidirectional encoder representations from transformers: Chinese named entity recognition using sequential lexicon enhanced BERT

Xin Liu[1], Jiashan Zhao[2], Junping Yao[1], Hao Zheng[1] and Zhong Wang[1]

[1] Department of Basic, Xi'an Research Institute of High-Tech, Xi'an, Shaanxi, China
[2] Department of Information and Network Management, Chang'an University, Xi'an, Shaanxi, China

## ABSTRACT

Lexicon Enhanced Bidirectional Encoder Representations from Transformers (LEBERT) has achieved great success in Chinese Named Entity Recognition (NER). LEBERT performs lexical enhancement with a Lexicon Adapter layer, which facilitates deep lexicon knowledge fusion at the lower layers of BERT. However, this method is likely to introduce noise words and does not consider the possible conflicts between words when fusing lexicon information. To address this issue, we advocate for a novel lexical enhancement method, Sequential Lexicon Enhanced BERT (SLEBERT) for the Chinese NER, which builds sequential lexicon to reduce noise words and resolve the problem of lexical conflict. Compared with LEBERT, it leverages the position encoding of sequential lexicon and adaptive attention mechanism of sequential lexicon to enhance the lexicon feature. Experiments on the four available datasets identified that SLEBERT outperforms other lexical enhancement models in performance and efficiency.

# INTRODUCTION

Named entity recognition (NER) is a significant task in natural language processing (NLP), aiming to recognize entities from texts and accurately identify their types. The kinds of these entities mainly include names of people, organizations, places, *etc.* NER often extracts the valuable information from the unstructured text, that can be used for many other high-level tasks, such as information retrieval, the construction of knowledge graphs, question-answering systems, public opinion analysis, biomedicine, recommendation systems, and so on. When compared with English NER (*Sun et al., 2020*), the Chinese NER is more difficult to process since it usually involves word segmentation. One intuitive way of performing Chinese NER (*Liu et al., 2022*) is to perform word segmentation first before applying word sequence labeling. However, incorrectly segmented entity boundaries in

Corresponding author
Jiashan Zhao, jszh@chd.edu.cn

segmentation lead to NER errors. To address this issue, most Chinese NER are character-based. Studies have shown that character-based methods outperform word-based methods for the Chinese NER, but character-based models still face the challenge of not completely extracting words and their sequence information. However, they are practically beneficial.

To overcome the problems, researchers began to explore the introduction of word information into character-based NER models, namely, lexical enhancement, to improve the performance of entity recognition. In recent years, researchers have proposed a variety of lexical enhancement methods (_Li et al., 2023_; _Li et al., 2024_; _Zhang et al., 2024_; _Zhang et al., 2023_). According to the different ways of introducing lexicon information, there are four lines of lexical enhancement methods for Chinese NER: dynamic architecture, adaptive embedding, graphical neural network, and pre-trained model. The dynamic architecture method suggests lexicon information by designing dynamic network structures, such as Lattice-long short-term memory (Lattice-LSTM; _Zhang & Yang, 2018_), lexicon rethinking convolutional neural network (LR-CNN; _Gui et al., 2019a_), and Flat-LAttice Transformer (FLAT; _Li et al., 2020_), and lightweight lexicon-enhanced transformer (LLET; _Ji & Xiao, 2024_). The adaptive embedding method presents an adaptive embedding representation of lexicon to represent entity information, like word-character (WC)-LSTM (_Liu et al., 2019_) and Simple-Lexicon (_Ma et al., 2019_), and span enhanced two-stage network with counterfactual rethinking (SENCR; _Zheng et al., 2024_). The graph neural network method uses graph structure to learn the representation of lexicon: collaborative graph network (CGN; _Sui et al., 2019_), lexicon-based graph neural network (LGN; _Gui et al., 2019b_), and randomly wired graph neural network (RWGNN; _Chen et al., 2023_). The pre-trained model method enhances the lexicon through the Bidirectional Encoder Representations from Transformers (BERT) pre-trained model, including BERT-BiLSTM-CRF (_Dai et al., 2019_) and Lexicon Enhanced Bidirectional Encoder Representations from Transformers (LEBERT) (_Liu et al., 2021_).

Although the LEBERT achieves relatively higher performance in multiple Chinese NER tasks, it has some limitations. First, the LEBERT introduce more potential words. In Fig. 1, "An City", "Chang'an", combined with lexical boundaries and contextual information, should be filtered. Second, the LEBERT cannot handle the problem of lexical information conflict. So, although the attention mechanism can assign different weights to different words, it does not consider the possible conflicts between words. In Fig. 1, "Xi'an City" and "Mayor" do not need to be introduced at the same time. Finally, the LEBERT does not distinguish the positional relationship between characters and words, resulting in insufficient utilization of word boundary information. For "Chang'an District", "Chang" is the first word, "An" is the middle word, and "District" is the last word.

To resolve these issues, a Chinese NER method is proposed based on sequential lexicon called Sequential Lexicon Enhanced BERT (SLEBERT). Sequential lexicon is a sequence of lexicon words that match a specific sentence. By constructing sequential lexicon, removing noise words, applying position encoding technology, and effectively overcoming the problem of lexical conflict by introducing an adaptive attention mechanism for the sequential lexicon is achieved. Adaptive attention mechanism will adjust attention weight adaptively based on improved loss function. Multiple experiments are conducted on four

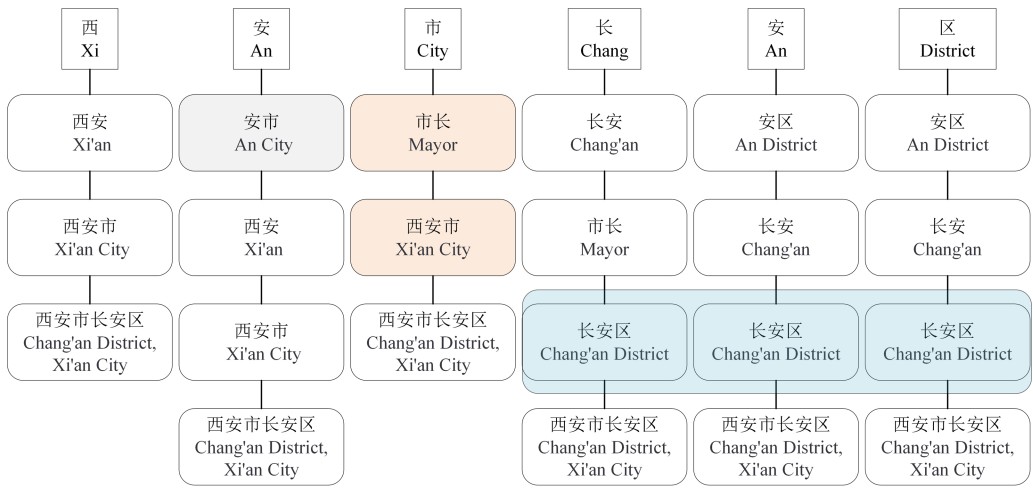

**Figure 1** **Character-word pairs in the LEBERT.**

available Chinese NER datasets, and the results verify the effectiveness of the proposed method.

The main contributions of the article are summarized as follows:

(1) A Chinese NER method based on sequential lexicon to enhance BERT is proposed to optimize LEBERT.

(2) The adaptive attention mechanism for the sequential lexicon is proposed to resolve the problem of lexicon information conflict.

(3) The four available datasets are applied to verify the effectiveness of the Chinese NER method based on sequential lexicon.

## RELATED WORK

Our work is in line with existing methods using neural network for Chinese NER (*Ahmad, Shah & Lee, 2023*; *Hu & Ma, 2023*; *Xue et al., 2023*). Character sequence labeling has been the dominant approach for Chinese NER, which has been empirically proven to be effective (*Ding et al., 2019*). One drawback of character-based NER, however, is that explicit word and word sequence information is not fully exploited, which can be potentially useful. To address this issue, *Zhang & Yang (2018)* first introduced lexicon into character-based NER model by using a lattice structure LSTM. Besides Lattice-LSTM, existing lexical enhancement methods also use many other network architectures, such as CNN in LR-CNN (*Gui et al., 2019a*), graph networks in CGN (*Sui et al., 2019*), LGN (*Gui et al., 2019b*) and the Transformer model in FLAT (*Li et al., 2020*), non-FLAT-lattice transformer (NFLAT; *Wu et al., 2022*). More recently, the popular pre-trained model has also been used for lexical enhancement, such as BERT-BiLSTM-CRF (*Dai et al., 2019*),

LEBERT (*Liu et al., 2021*) is in line with our work, focusing on lexical enhancement. Experimental results show that LEBERT achieves state-of-art performance which directly injects lexicon information between Transformer layers in BERT using a Lexicon Adapter.

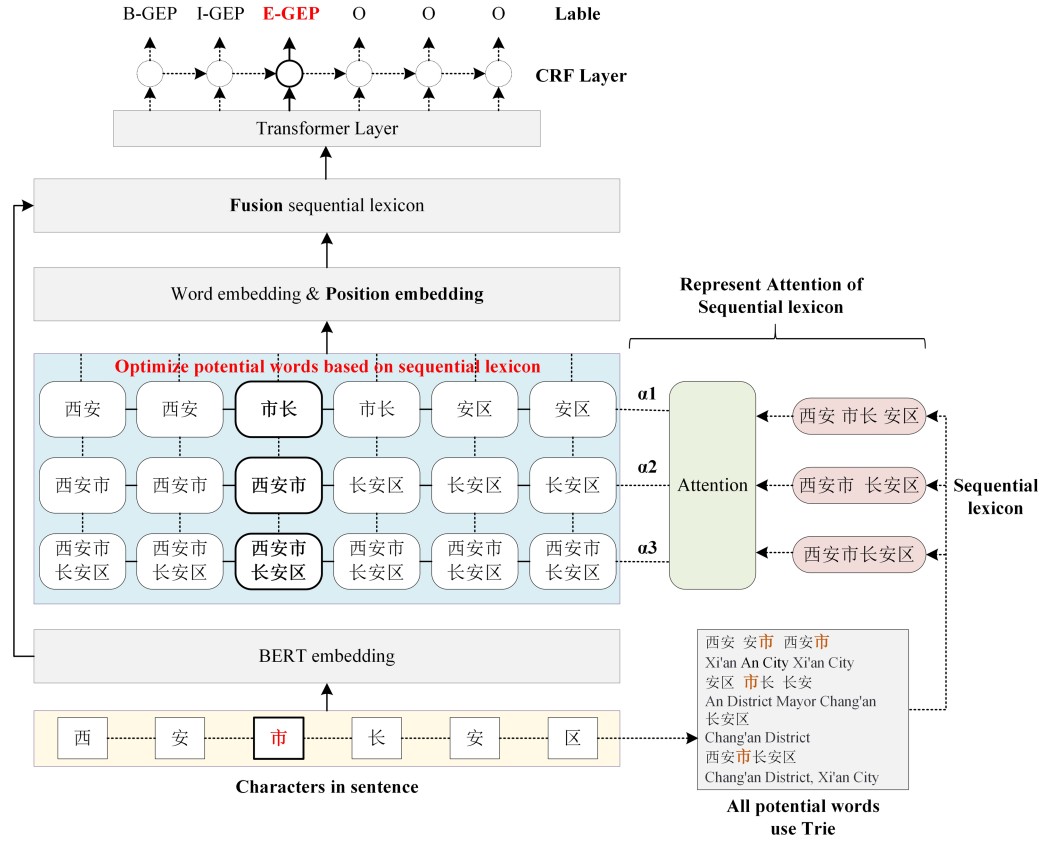

**Figure 2** The architecture of sequential lexicon enhanced BERT.

However, this method is likely to introduce noise words and does not consider the possible conflicts between words when fusing lexicon information.

## METHODS

The main architecture of the proposed Sequential Lexicon Enhanced BERT is shown in Fig. 2. The SLEBERT has three main differences when compared to LEBERT. First, the SLEBERT improves the quality of potential words by creating a sequential lexicon, which contributes to the elimination of interference words and enhances relative lexicon information. Second, position encoding and attention to sequential lexicon are used to fuse the lexical feature into BERT. Finally, the adaptive attention mechanism for the sequential lexicon is suggested to resolve the problem of lexical conflict.

In this section we describe: (1) Sequential Lexicon Building ('Sequential Lexicon Building'), which improves the quality of potential words; (2) Sequential Lexicon Fusion ('Sequential Lexicon Fusion'), by injecting position embedding and representing attention of sequential lexicon; (3) Adaptive Attention ('Adaptive Attention'), which resolves the problem of lexical conflict.

The succeeding sections present the construction of a sequential lexicon, which improves the quality of potential words; the fusion of sequential lexicon by plugging position

embedding and attention of sequential lexicon; and the implementation of adaptive attention, which resolves the problem of lexical conflict.

## Sequential lexicon building

For Chinese NER, character-based NER methods usually represent a Chinese sentence as a character sequence, containing character-level features solely. To make use of lexicon information, recent work tries to integrate the lexicon to enhance character-based models (*Ji & Xiao, 2024*; *Xiao, Ji & Li, 2024*; *Zhang et al., 2024*). *Liu et al. (2021)* introduced LEBERT to integrate lexicon information between Transformer layers of BERT directly, which showed good performance. However, LEBERT still has some challenges. For example, it introduces interference words and does not consider the possible conflicts between words when fusing lexicon information. To address these issues, we propose a novel lexical enhancement method, SLEBERT, that uses Sequential Lexicon to optimize LEBERT. Specifically, we first build a Trie based on a Chinese Lexicon, and use the Trie to obtain all of the potential words by traversing each character in a sentence instead of word segmentation. As shown in Fig. 1, for the character "安(An)", we can find out four potential words, namely "安市(An City)", "西安(Xi'an)", "西安市(Xi'an City)", and "西安市长安区(Chang'an District, Xi'an City)". Then, we build a sequential lexicon to reduces noise words and improve the quality of potential words. The specific process of building a sequential lexicon can be described by Algorithm 1.

---

**Algorithm 1:** To build sequential lexicon

**Input**: Lexicon Trie T, the sequential character of Chinese sentence $C = (C_1, C_2, \ldots, C_n)$

**Output**: The k-th sequential lexicon $LS = (LS_1, LS_2, \ldots, LS_k)$

1. Initialize an empty list LC.
2. For each character c in C, do: build the first word list append to LC.
3. Initialize an empty list LS.
4. For each character c in C, do: use depth search algorithm DFS(index, current_sequence) to build sequential lexicon LS.

---

Inferences drawn from the built sequential lexcion indicate that sequential lexicon requires optimization. For example, in Fig. 3, the sequential lexicon (Xi'an, mayor, Chang'an District) contains the sequential lexicon (mayor, Chang'an District). In addition, when compared with the sequential lexicon (Xi'an, mayor, Chang'an), the sequential lexicon (Xi'an, mayor, Chang'an district) is more accurate. Then we calculate the coverage rate of each sequential lexicon, as shown in Eq. (1), to optimize the sequential lexicon building.

Where $m$ represents the total number of words in the set $LC_i$, $CV_i$ indicates the coverage rate of the $i$th sequence. The sequential lexicon are finally optimized into the top groups

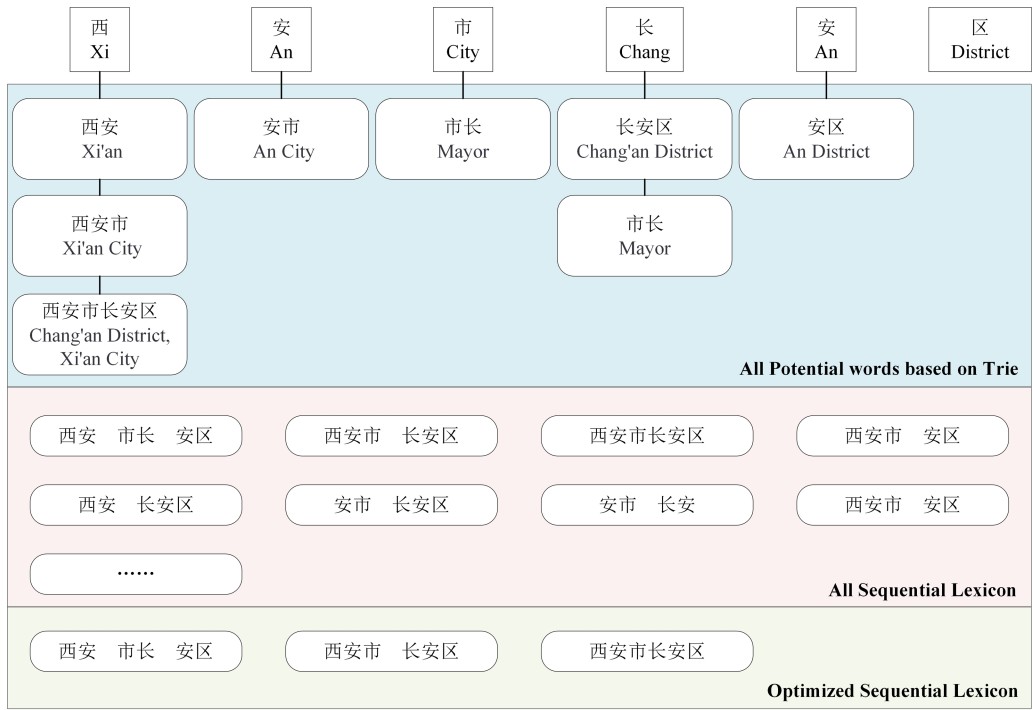

**Figure 3   An example of word sequence building and its optimization.**

based on coverage rate.

$$CV_i = \frac{\sum_{j=0}^{m} len(lc_{ij})}{len(S)} \tag{1}$$

## Sequential lexicon fusion

The sequential lexicon is finally integrated into BERT in the form of the lexicon. First, extract the potential words corresponding to each character of a Chinese sentence based on sequential lexicon. Second, employ the attention of sequence lexicon to combine the potential words with the corresponding character, as shown in Fig. 4, which differs from the attention of lexicon employed in LEBERT. Finally, we use position encoding and attention of sequential lexicon to infuse the lexical feature into BERT.

### Extraction of potential words

The potential words corresponding to each character of a Chinese sentence are attained based on the sequential lexicon. The specific method can be described by Algorithm 2. $LX_i$ represents the potential words set corresponding to the $i$th character. $LX_{i,j}$ characterizes the potential words of the $i$th character obtained based on the $j$th sequential lexicon.

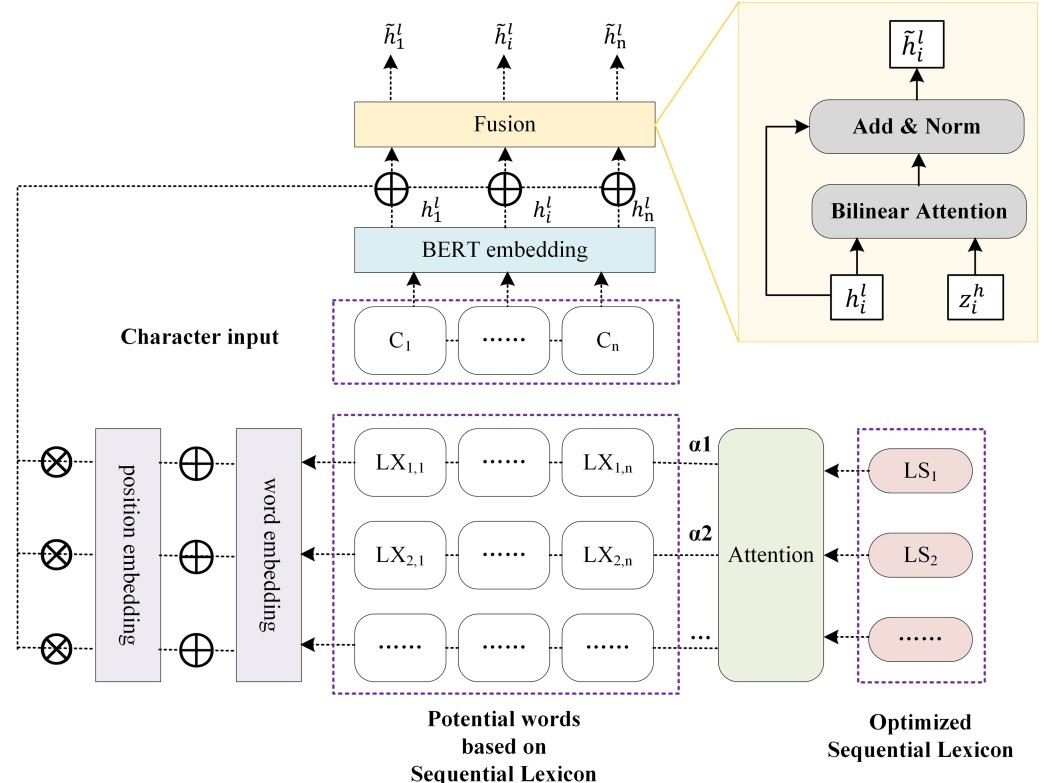

**Figure 4** The enhancement using sequential lexicon.

**Algorithm 2:** To extract the potential words based on sequential lexicon

**Input**: Sequential lexicon of TOP groups $LS = (LS_1, LS_2, \ldots, LS_{TOP})$

**Output**: The potential words set LX, a two-dimensional array

1. For each lexicon in the top group of sequential lexicon, get the start position and end position of each lexicon.

2. For each lexicon in the top group of sequential lexicon, assign the lexicon to the LX matrix at the position[i][j], where i is the position of character in sentence and j is the index of lexicon sequential group.

### *Representation of position encoding*

Considering that the boundary contributions in different positions of the lexicon are different, position encoding is applied to add in embedding layer. The $i$th words in $j$th sequential lexicon group is represented as following:

$$x_{ij} = e^w \left( LX_{ij} \right) \tag{2}$$

$$p_i = \cos\left(i/1000^{2i/d_e}\right), q_i = \sin\left(i/1000^{2i/d_e}\right), P_i = p_i + q_i \tag{3}$$

$$\acute{x}_{ij} = x_{ij} + P_i \tag{4}$$

where $e^w$ is a pre-trained word embedding lookup table, $d_e$ denotes the dimension of word embedding, $i$ represents the $i$th word as well as the position in sequential lexicon, $P_i$ is position encoding, which is a combination of sine and cosine functions, and $\acute{x}_{ij}$ is word embedding fusing position encoding.

### *Fusion with attention of sequential lexicon*

Each character is paired with multiple potential words, and each potential word corresponds to a sequential lexicon group. The contributions of distinct sequential lexicon are different. For example, for "Chang'an District, Xi'an City", the sequential lexicon "Xi'an City, Chang'an District" should contribute the most because it is closest to the true segmentation. To pick the most relevant words in the lexicon set, the attention mechanism is introduced here.

Considering that sequential lexicon contains lexicon, the attention of sequential lexicon is therefore used. Specifically, $A = (a_1, a_2, \ldots, a_{TOP})$ represents the attention vector, $a_j$ represents the attention of the $j$th sequential lexicon group, it can be calculated as:

$$s_j = \frac{1}{n}\sum_{i=1}^{n}\acute{x}_{ij}, Q = s_j W_Q, K = s_j W_K \qquad (5)$$

$$a_j = softmax\left(\frac{QK^T}{\sqrt{d_e}}\right). \qquad (6)$$

Consequently, we can get the weighted sum of all words by:

$$z_i^h = \sum_{j=1}^{TOP} a_j v_{ij}^h \qquad (7)$$

$$v_{ij}^h = W_2\left(\tanh\left(W_1 \acute{x}_{ij} + b_1\right)\right) + b_2 \qquad (8)$$

where $v_i^h$ is the hidden vectors of each word, $W_1$ and $W_2$ denote matrices of dimensions $d_h \times d_e$ and $d_h \times d_h$, respectively, and $d_h$ represents the size of the BERT hidden layer.

$H^l = \left(h_1^l, h_2^l, \ldots, h_n^l\right)$ represents the output of the Transformer from layer $l$ in the BERT model, $\tilde{h}_i^l$ represents the vector output of layer $l$ after integrating lexicon information. Finally, the lexicon information is injected into the character vector by:

$$\tilde{h}_i^l = h_i^l + z_i^h \qquad (9)$$

### Adaptive attention

Considering the possible conflicts between words, as shown in Fig. 5, between "Xi'an city" and "Mayor", if one of the words is close to the real semantic meaning, that is, the enhancement attention of the word is high and the attention of the other word should be very low.

Therefore, the sequential lexicon's adaptive attention mechanism is proposed to resolve the problem of lexical conflict by incorporating an extra module M into the L2 loss function,

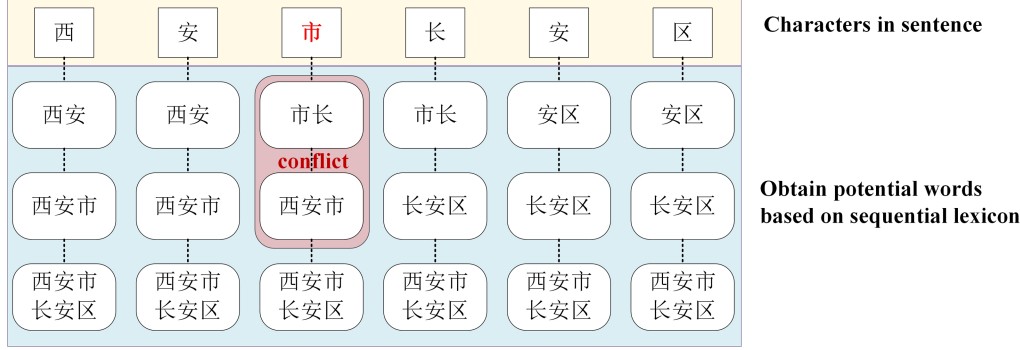

**Figure 5** An example of lexical enhancement conflict.

delineated by Eq. (10). Thus, the attention mechanism is adjusted adaptively.

$$L = -\sum_{i=1}^{n} log\left(p\left(y_i|c_i\right)\right) + \lambda_1 M + \frac{\lambda_2}{2}\left(\|\theta\|^2\right) \tag{10}$$

where $\lambda_1$ is a weight parameter, responsible for the extra module. $\lambda_2$ denotes the regularization parameter of $L_2$, $\theta$ is the set of all trainable parameters. The extra module $M$ is defined by:

$$M = -log\left(\sum_{i=1}^{n} f_i\right) \tag{11}$$

$$f_i = \left(u_j \sum_{j=1}^{k} o_{ij} a_j\right)^2, u_i = \begin{cases} 0, when & t_i \neq 0 \\ 1, when & t_i = 0, t_i = \sum_{j=1}^{k} o_{ij} \end{cases} \tag{12}$$

$$o_{ij} = \begin{cases} 1, & when\ c_i\ is\ the\ first\ character\ of\ LX_{ij} \\ -1, & when\ c_i\ is\ the\ last\ character\ of\ LX_{ij} \\ 0, & others \end{cases} \tag{13}$$

where $f_i$ denotes the lexical conflict reflection function. $a_j$ designates the attention of the $j$th sequential lexicon group. $o_{ij}$ indicates whether the $i$th character is the first character or the last character of the $j$th potential words corresponding to the $i$th character. $u_j$ indicates whether there is lexical conflict in the $i$th character. It is judged by calculation $t_j$, if there exists the $i$th character as both the first character and the last character in the potential words set, that means there is lexical conflict.

# EXPERIMENTS

We evaluate the proposed SLEBERT method using the F1 score (F1), precision (P), and recall (R) metrics, with a comparison of several lexical enhancement models. At the same

**Table 1  Public datasets of Chinese NER.**

| Corpus | Entity types | URL |
|---|---|---|
| Weibo NER | Person, Location, Organization and Geo-political | https://huggingface.co/datasets/hltcoe/weibo_ner |
| MSRA | Person, Location, Organization | https://github.com/InsaneLife/ChineseNLPCorpus/tree/master/NER/MSRA |
| Resume NER | Person, Location, Organization, Country, Education, Profession, Race, Title | https://github.com/GuocaiL/nlp_corpus/tree/main/open_ner_data/ResumeNER |
| OntoNotes Release 5.0 | Preson, NORP, Facility, Organization, GPE, Location, Product, Event, Work of art, Law, Language, Date, Time, Percent, Money, Quantity, Ordinal, Cardinal | https://doi.org/10.35111/xmhb-2b84 |

**Table 2  The statistics of the datasets.**

| Datasets | Type | Train | Dev | Test |
|---|---|---|---|---|
| OntoNotes | Sentences | 15.7k | 4.3k | 4.3k |
| | Char | 491.9k | 200.5k | 208.1k |
| MSRA | Sentence | 46.4k | 4.4k | 4.4k |
| | Char | 2169.9k | 172.6k | 172.6k |
| Weibo | Sentence | 1.4k | 0.27 k | 0.27 k |
| | Char | 73.8k | 14.5k | 14.8k |
| Resume | Sentence | 3.8k | 0.46k | 0.48k |
| | Char | 124.1k | 13.9k | 15.1k |

time, this section performs ablation experiments to verify the effectiveness of the proposed sequential lexicon Chinese NER model. Last, the advantages of implementing sequential lexicon are analyzed.

### Datasets

We evaluate the performance of SLEBERT on four mainstream Chinese NER benchmarking datasets, including Weibo NER (*Peng & Dredze, 2015*; *Peng & Dredze, 2016*; *Weischedel et al., 2011*), OntoNotes (*Weischedel et al., 2011*), Resume NER (*Yu et al., 2022*), and MSRA (*Zhu, Wang & Karlsson, 2019*). Weibo NER is a social media domain dataset, which is drawn from Sina Weibo; while OntoNotes and MSRA datasets are in the news domain. Resume NER dataset consists of resumes of senior executives. Table 1 lists the entity types of each dataset. Table 2 provides a detailed overview of each dataset, including the sizes of the training, development, and test sets.

### Experimental setting

Our model is constructed based on LEBERT, using the 200-dimension pre-trained word embedding from *Song et al. (2018)*, the Adam optimizer with an initial learning rate of 1e−5 for the original parameters of BERT, 1e−4 for other parameters introduced by SLEBERT, and a maximum epoch number of 20 for training on all datasets. As shown in Fig. 6, the results on Resume NER, OntoNotes, and MSRA become more stable with increasing training iterations. Table 3 shows the maximum and average character lengths

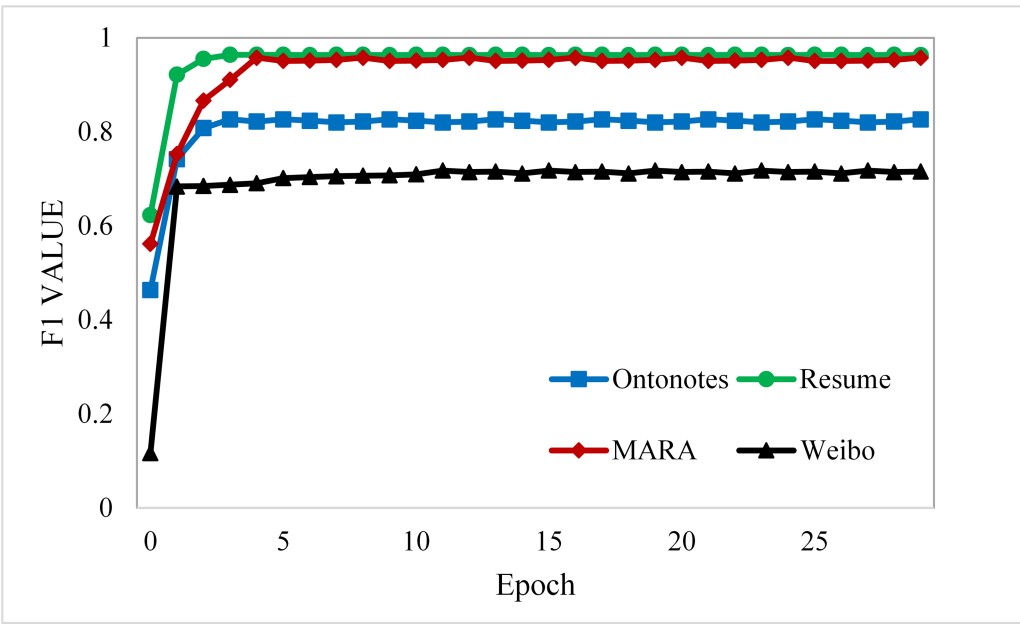

**Figure 6**  Learning curves for the four datasets.

**Table 3**  The maximum and average character lengths for each dataset.

| Datasets | The maximum character length | The average character length |
|---|---|---|
| Weibo NER | 175 | 54 |
| Resume NER | 178 | 42 |
| OntoNotes | 273 | 37 |
| MSRA | 281 | 46 |

for each dataset sample and we set the max length of the sequence to 256; the training batch size is 20 for MSRA and four for the other datasets.

## Experimental results

To evaluate the effectiveness of the proposed SLEBERT, we compare our SLEBERT method with almost all the lexical enhancement models, including Lattice-LSTM, LR-CNN, CGN, FLAT, and LEBERT. These models achieve superior performance from various network architectures. In Table 4, we can see that SLEBERT improves the performance of LEBERT. The overall F1 score on Weibo NER is increased by 1.01%. The F1 score on OntoNotes is increased by 0.6%. The F1 score obtained on Resume NER is increased by 0.3%. For the MSRA dataset, the proposed method increases the F1 score by 0.07%.

## How sequential lexicon performs

To verify the effectiveness of sequential lexicon, we perform two ablation experiments. First, we remove the adaptive attention mechanism, and denote this method as 'SLEBERT-'. Second, we remove the adaptive attention mechanism and position encoding, which merely

**Table 4  NER result.**

| Models | OntoNotes | | | MARA | | | Weibo NER | | | Resume NER | | |
|---|---|---|---|---|---|---|---|---|---|---|---|---|
| | P | R | F | P | R | F | P | R | F | P | R | F |
| Lattice -LSTM | 76.35 | 71.56 | 73.88 | 93.57 | 92.79 | 93.18 | 53.04 | 62.25 | 58.79 | 94.81 | 94.11 | 94.46 |
| LR-CNN | 76.4 | 72.6 | 74.45 | 94.5 | 92.93 | 93.71 | 57.14 | 66.67 | 59.92 | 95.37 | 94.84 | 95.11 |
| CGN | 76.27 | 72.74 | 74.46 | 94.01 | 92.93 | 93.63 | 56.45 | 68.32 | 65.18 | 94.27 | 94.59 | 94.43 |
| FLAT | – | – | 75.7 | – | – | 94.35 | – | – | 63.42 | – | – | 95.45 |
| LEBERT | – | – | 82.08 | – | – | 95.7 | – | – | 70.75 | – | – | 96.08 |
| SLEBERT | 82.84 | 82.53 | 82.68 | 95.88 | 95.67 | 95.78 | 73.83 | 69.81 | 71.76 | 96.44 | 96.32 | 96.38 |

**Table 5  Ablation experimental results.**

| Models | OntoNotes | | | MARA | | | Weibo NER | | | Resume NER | | |
|---|---|---|---|---|---|---|---|---|---|---|---|---|
| | P | R | F | P | R | F | P | R | F | P | R | F |
| LEBERT | – | – | 82.08 | – | – | 95.7 | – | – | 70.75 | – | – | 96.08 |
| SLEBERT | 82.84 | 82.53 | 82.68 | 95.88 | 95.67 | 95.88 | 73.83 | 69.81 | 71.76 | 96.44 | 96.32 | 96.38 |
| SLEBERT- | 82.84 | 82.15 | 82.49 | 95.88 | 95.67 | 95.88 | 73.83 | 68.54 | 71.09 | 96.39 | 96.02 | 96.21 |
| SLEBERT-- | 82.84 | 81.84 | 82.34 | 95.7 | 95.67 | 95.7 | 73.83 | 68.32 | 70.97 | 96.39 | 96.02 | 96.21 |

utilize sequential lexicon to improve the quality of potential words. We denote this method as 'SLEBERT–'. The results of the ablation experiments are reported in Table 5.

Based on the ablation experimental results, we can see that the performance of SLEBERT improved when we merely utilized sequential lexicon as compared with LEBERT. This is because sequential lexicon reduces noise words that can potentially confuse NER.

The input layer in BERT already takes into account the position encoding, what differs is that SLEBERT considers the position of a word within the sequential lexicon, but not the position of a character within input sentences, so the performance is only marginally improved when add relative position encoding.

We can also see that the performance of SLEBERT degrades when removing the adaptive attention mechanism. The adaptive attention module resolves the problem of lexical conflict.

Briefly, sequential lexicon reduces noise words, position encoding distinguishes words from different sequences, and adaptive attention mechanism extracts the most relevant words.

## Advantages of using sequential lexicon
### F1 against sentence length
Figure 7 shows the F1 value trend of the LEBERT and SLEBERT on the OntoNotes dataset. They show a similar performance-length curve, decreasing as the sentence length increases. We speculate that for lexicon enhanced models it may become more difficult to exact correct potential words because of the increased number of potential words as the sentence

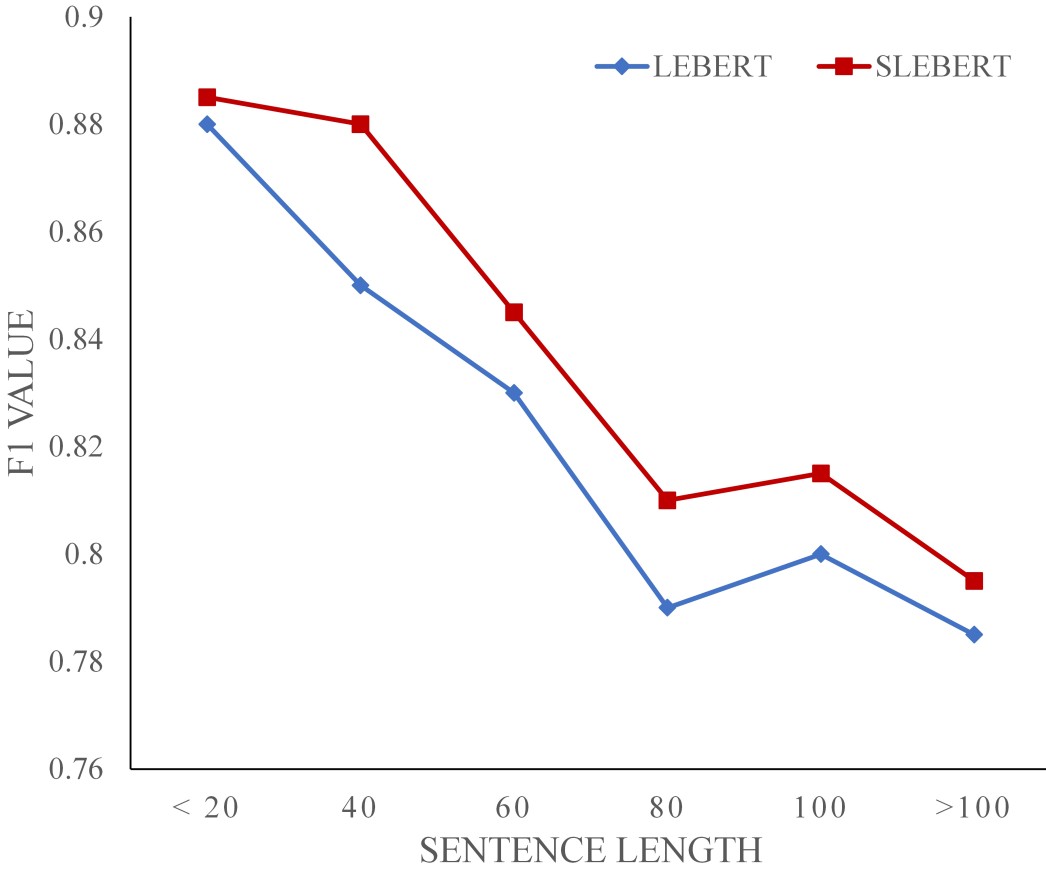

**Figure 7** **F1 value against the sentence length.**

become longer. Compared with LEBERT, SLEBERT performs better when sentence length increases, demonstrating the more effective use of sequential lexicon.

### Decrease rate of lexical conflict

Figure 8 shows the lexical conflict percentage between LEBERT and SLEBERT. We can see that sequential lexicon is useful for reducing noise words, such as decreasing the conflict words in the Train Dataset from 20.57% to 2.80% on OntoNotes, from 18.18% to 1.89% on MSRA, from 19.03% to 1.88% on Weibo NER, from 9.36% to 3.01% on Resume NER. On the other hand, the LEBERT indeed introduces many conflicts words that may affect the accuracy of Chinese NER and SLEBERT can improve the performance by sequential lexicon.

### Case study

Table 6 shows several case studies comparing LEBERT and SLEBERT. It can be seen there are many irrelevant words from the LEBERT potential words. In sentence 1, "子小 (Zi Xiao)" and "小菇 (Xiao Gu)" is not valid and introduce confusion for person entity "小菇 凉 (Little Girl)", but SLEBERT can correctly extract "小菇凉 (Little Girl)". In sentence 2, "华克 (Hua Ke)"and "克山 (Ke Shan)" is not valid and introduce confusion for location

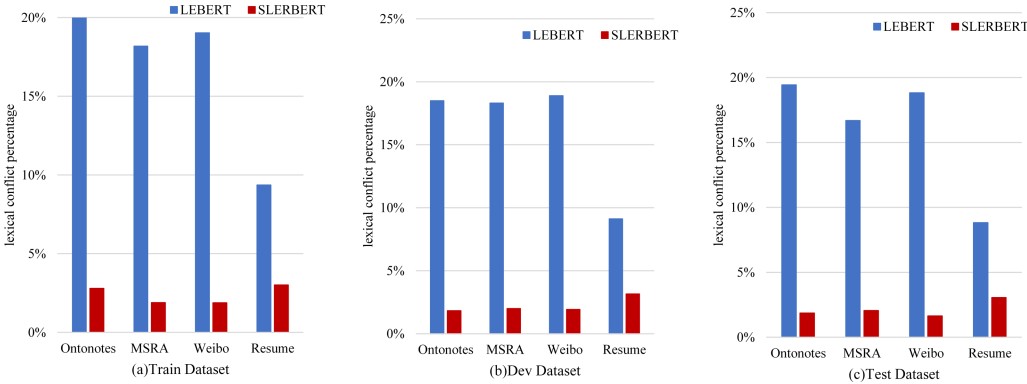

**Figure 8** A comparison between LEBERT and SLEBERT in terms of efficiency, evaluated at lexical conflict percentage.

**Table 6** Examples of potential words.

| | |
|---|---|
| sentence1(truncated) | 雯子小菇凉(Wen Zi Little Girl) |
| characters | 雯 子 小 菇 凉 |
| labels | B-PER.NAM I-PER.NAM I-PER.NOM I-PER.NOM I-PER.NOM |
| LEBERT<br><br>potential words | [雯子], [雯子,子小], [子小,小菇,小菇凉], [小菇,小菇凉,菇凉], [小菇凉,菇凉,凉] |
| SLEBERT<br>（TOP=1）<br>potential words | [雯子], [雯子], [雯子], [小菇凉], [小菇凉], [小菇凉] |
| sentence1(truncated) | 华克山庄免税店(Walker Hill Duty-Free Shop) |
| characters | 华 克 山 庄 免 税 店 |
| labels | B-LOC.NAM I-LOC.NAM I-LOC.NAM I-LOC.NAM B-ORG.NOM I-ORG.NOM I-ORG.NOM |
| LEBERT<br><br>potential words | [华克,华克山庄], [华克,华克山庄,克山], [华克山庄,克山,山庄], [华克山,山庄,庄],<br><br>[免税,免税店], [免税,免税店,税], [免税店,店] |
| SLEBERT<br>（TOP=1）<br>potential words | [华克山庄], [华克山庄], [华克山庄], [华克山庄],<br><br>[免税店], [免税店], [免税店] |

entity "华克山庄 (Walker Hill)", but SLEBERT can correctly extract "华克山庄 (Walker Hill)". The above suggests that SLEBERT can reduce noise words and extract more effective potential words by sequential lexicon.

## CONCLUSIONS

We presented a novel lexical enhancement method, Sequential Lexicon Enhanced BERT for Chinese NER, which builds sequential lexicon to reduces noise words and resolve the problem of lexical conflict, finding that it gives good performance compared to LEBERT. The SLEBERT model is more effective in lexical enhancement thanks to the position encoding of sequential lexicon and adaptive attention mechanism of sequential lexicon. The performance of our method is limited by the quality and domain of the lexicon. These factors will affect the performance of NER to varying degrees. In addition, the sequential lexicon group may affect the accuracy of our NER model since the value of TOP when optimizing sequential lexicon affects the number of potential words. We leave the investigation of such influence to future work.

## ACKNOWLEDGEMENTS

We thank Hao Zheng and Jiashan Zhao for their achievements that inspired this work, and we would like to thank the anonymous reviewers for their helpful remarks.

### Funding

This work was supported by Xi'an Research Institute of High-Tech Fund (No. 2022QN-S008). The funders had no role in study design, data collection and analysis, decision to publish, or preparation of the manuscript.

### Grant Disclosures

The following grant information was disclosed by the authors:
Xi'an Research Institute of High-Tech Fund: No. 2022QN-S008.

### Competing Interests

The authors declare there are no competing interests.

### Author Contributions

- Xin Liu conceived and designed the experiments, performed the experiments, analyzed the data, performed the computation work, prepared figures and/or tables, authored or reviewed drafts of the article, and approved the final draft.
- Jiashan Zhao conceived and designed the experiments, performed the experiments, analyzed the data, authored or reviewed drafts of the article, and approved the final draft.
- Junping Yao conceived and designed the experiments, authored or reviewed drafts of the article, and approved the final draft.
- Hao Zheng conceived and designed the experiments, analyzed the data, authored or reviewed drafts of the article, and approved the final draft.
- Zhong Wang conceived and designed the experiments, analyzed the data, authored or reviewed drafts of the article, and approved the final draft.

## Data Availability

The code and data are available in the Supplementary Files.

The third-party data are available at:

- Weibo NER: https://huggingface.co/datasets/hltcoe/weibo_ner
- MSRA: https://github.com/InsaneLife/ChineseNLPCorpus/tree/master/NER/MSRA
- Resume NER: https://github.com/GuocaiL/nlp_corpus/tree/main/open_ner_data/ResumeNER
- OntoNotes Release 5.0: https://doi.org/10.35111/xmhb-2b84.

## Supplemental Information

Supplemental information for this article can be found online at http://dx.doi.org/10.7717/peerj-cs.2344#supplemental-information.

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
