# Peer review of "Sequential lexicon enhanced bidirectional encoder representations from transformers: Chinese named entity recognition using sequential lexicon enhanced BERT"

_PeerJ Computer Science, doi:10.7717/peerj-cs.2344_

## Round 0.1 · original submission · Minor Revisions

Thank you for submitting your manuscript to PeerJ Computer Science. Based on feedback from three reviewers, we invite you to revise your manuscript. The primary concerns include the disclosure of experimental settings and data processing details. Additional information on the methods may be required to ensure reproducibility. Also, the grammar mistakes need to be addressed. Please feel free to contact us if you have any questions.

Reviewer 1 ·

Basic reporting

The paper presents a significant advancement in Chinese NER through the development of SLEBERT, demonstrating improved performance and robustness over existing models. Here are the overall assessment of the paper:

Clarity and Structure: The manuscript is well-structured and clearly written, with a logical flow from the introduction to the conclusion. Each section is well-defined, providing a clear understanding of the research problem, methodology, experiments, and findings.

Literature Review: The paper includes a comprehensive literature review, discussing various approaches to Chinese NER and lexical enhancement methods. However, it could benefit from a deeper comparison with the latest state-of-the-art techniques to highlight the novelty of SLEBERT more explicitly.

Figures and Tables: The figures and tables are relevant and clearly presented. They effectively illustrate the performance improvements and comparative analyses. However, some additional visual aids could further enhance understanding, particularly for the architectural components of SLEBERT.

References: The references are relevant and up-to-date, providing a solid foundation for the study. However, there is some inconsistency in the citation style that should be corrected for uniformity.

Experimental design

Methodology: The methodology is robust and well-detailed. The sequential lexicon building, lexicon fusion, and adaptive attention mechanisms are well-explained. The use of four benchmark datasets provides a comprehensive evaluation of the proposed method.

Experimental Setup: The experimental setup, including the description of datasets, training parameters, and evaluation metrics, is clearly outlined. This transparency allows for reproducibility and verification of results.

Comparative Analysis: The comparative analysis with existing models like Lattice-LSTM, LR-CNN, CGN, FLAT, and LEBERT is thorough. However, it would be beneficial to include a more detailed discussion on the choice of these specific models for comparison.

Validity of the findings

Effectiveness of Sequential Lexicon: The paper convincingly demonstrates that the sequential lexicon enhances the quality of potential words and reduces noise. The results show a clear improvement in F1 scores across all datasets. However, additional experiments on more diverse datasets would strengthen the generalizability of the findings.

Adaptive Attention Mechanism: The adaptive attention mechanism is shown to effectively resolve lexical conflicts, improving performance. A more detailed analysis of how this mechanism impacts different types of entities (e.g., person, location, organization) would provide deeper insights into its effectiveness.

Ablation Studies: The ablation studies are well-conducted, showing the contributions of different components of SLEBERT. However, a more granular analysis of the impact of each component on various performance metrics (precision, recall) would be beneficial.

Additional comments

Terminology Consistency: Ensure consistent use of terminology throughout the paper. For example, the terms "sequential lexicon" and "lexicon sequence" seem to be used interchangeably but should be clearly defined and used consistently.

Algorithm Descriptions: The algorithms presented (e.g., Algorithm 1 and 2) are clear, but including pseudocode or flowcharts could enhance understanding.

Hyperparameter Details: Provide more details on the hyperparameter tuning process. This information would be valuable for researchers attempting to replicate or build upon this work.

Error Analysis: Include a more detailed error analysis, particularly focusing on cases where SLEBERT underperforms compared to other models. Understanding these limitations could provide directions for future improvements.

Language and Grammar: The manuscript is generally well-written, but a thorough proofread could eliminate minor grammatical errors and improve overall readability.

Reviewer 2 ·

Basic reporting

1. The manuscript uses clear and professional English throughout. However, there are sections where the language is overly verbose and could benefit from more concise phrasing.
2. The literature review is comprehensive and provides sufficient background context. The authors use references that are recent.
3. The manuscript follows a professional structure with well-organized figures and tables. The raw data is shared as per the journal's policy.
4. The manuscript is self-contained, presenting relevant results that address the research hypotheses. It also uses clear definitions and descriptions of the terms and methods used. The mathematical formulations are clear.

Experimental design

1. The manuscript does not justify the choice of certain experimental settings, such as the learning rate, epoch number, and batch size. The authors should justify the choice of experimental settings by providing reasoning based on preliminary experiments, literature references, or theoretical considerations. Explain why these specific values were chosen and how they contribute to the performance of the model.
2. The manuscript lacks a detailed description of the data preprocessing steps. This includes how the raw text data was cleaned, tokenized, and transformed into a suitable format for the model. The authors should consider providing a description either in the main text or in the appendix.
3. The author could also provide some details on data preprocessing. Describe the steps taken to clean, tokenize, and preprocess the text data.

Validity of the findings

1. The authors provide a thorough analysis of the methodologies; however, the potential implications of this method are lacking.
2. Please also consider providing more details on the case study.
3. Limitations and future works of this research is not provided.

Additional comments

NA

Reviewer 3 ·

Basic reporting

Liu et al. proposed a sequential lexicon enhanced BERT (SLEBERT) for Chinese Named Entity Recognition (NER). This work addressed the limitation of lexicon information conflict in current lexicon enhanced BERT methods by introducing sequential lexicon, accounting for positional relationship between characters and words, and using adaptive attention to resolve lexical conflict. This new approach effectively improves Chinese NER by substantially reducing lexical conflict, which can provide more precise recognition of Chinese words given the sentence. Overall, the authors did nice job presenting their algorithms and comparing with other existing approaches.

Experimental design

1. In real data experiments, can the authors elaborate on why the performance metric (F1) differs between four datasets? For example, resume texts tend to be in a more structured format and short sentence, whereas the others are less organized and complex to disassociate.
2. Table 1 WeiBo data set URL broke.

Validity of the findings

1. Can the authors show evidence for line 267-268: “Compared with LEBERT, SLEBERT performs better and shows more robustness when sentence length increases”. It looks like the decreasing relationship is similar between these two methods as shown in Figure 6.

2. Typo in line 94: it should be “state-of-art”.

---

## Round 0.2 · Minor Revisions

Thank you so much for submitting the responses. One of the reviewers highlighted several minor points that have to be addressed to improve the manuscript for some plots and texts/wordings. Please feel free to let us know if you have any questions in the meantime.

Reviewer 1 ·

Basic reporting

The manuscript presents a novel approach, SLEBERT (Sequential Lexicon Enhanced BERT), for improving Chinese Named Entity Recognition (NER). The study addresses limitations in previous models by reducing noise words and resolving lexical conflicts through the use of a sequential lexicon and adaptive attention mechanisms. The research is original, well-structured, and within the journal's scope. While the experimental design is robust and the findings are valid, some improvements in language clarity, figure quality, and the inclusion of more recent literature references would enhance the manuscript. Overall, the study makes a meaningful contribution to the field of Chinese NER.

Clear and unambiguous, professional English used throughout.
The manuscript is generally clear, but there are some instances where the language could be more concise and less repetitive. For example, the introduction section contains sentences that could be simplified for better readability. The term "sequential lexicon" is repeatedly used without clear initial definition, which might confuse readers unfamiliar with the concept. I recommend defining key terms more clearly at the beginning of the manuscript and refining the language to be more concise. Here are the point by point review opinions:

1. Literature references, sufficient field background/context provided.
The manuscript provides a good overview of related work and references relevant literature adequately. However, some references could be updated to include more recent works, particularly in the area of neural network models and named entity recognition (NER) for Chinese. Expanding the related work section to discuss recent advancements in the field could strengthen the manuscript.

2. Professional article structure, figures, tables. Raw data shared.
The structure of the article is appropriate and follows the journal’s standards. Figures and tables are relevant, well-labeled, and support the text effectively. However, the quality of some figures, such as Figures 2 and 5, could be improved for better clarity. The raw data associated with the experiments have been shared, which is commendable.

3. Self-contained with relevant results to hypotheses.
The manuscript is self-contained and presents relevant results that support the stated hypotheses. The results section is thorough, but it could benefit from a more detailed explanation of how the results directly support the hypotheses.

4. Formal results should include clear definitions of all terms and theorems, and detailed proofs.
No formal theorems or proofs are provided or required in this context. However, the definitions of key terms, such as "sequential lexicon" and "adaptive attention mechanism," should be clarified early in the manuscript to enhance understanding.

Experimental design

1. Original primary research within Aims and Scope of the journal.
The research is original and falls within the scope of the journal. It introduces a novel method for Chinese NER using a sequential lexicon enhanced BERT model, which is a relevant and meaningful contribution to the field.

2. Research question well defined, relevant & meaningful. It is stated how research fills an identified knowledge gap.
The research question is well defined, focusing on improving the performance of Chinese NER models by addressing the limitations of existing methods. The manuscript clearly states the knowledge gap related to the introduction of noise words and lexical conflicts in prior models and proposes a solution through the sequential lexicon approach.

3. Rigorous investigation performed to a high technical & ethical standard.
The investigation appears to be rigorous, with a well-designed experimental setup. The authors have conducted thorough experiments across multiple datasets, which supports the robustness of their findings. Ethical standards seem to be upheld, although this aspect is not explicitly discussed in the manuscript.

4. Methods described with sufficient detail & information to replicate.
The methods are described in detail, allowing for replication. The algorithms are presented clearly, and the process of building and using the sequential lexicon is well explained. However, including pseudo-code or more detailed flow diagrams could further enhance the clarity of the methodology.

Validity of the findings

1. Impact and novelty not assessed. Meaningful replication encouraged where rationale & benefit to literature is clearly stated.
The impact and novelty of the research are evident in the proposed method and its improvements over existing models. The manuscript encourages meaningful replication by providing detailed descriptions and sharing the raw data. The rationale and benefit to the literature are clearly articulated.

2. All underlying data have been provided; they are robust, statistically sound, & controlled.
The underlying data are provided and appear robust. The statistical analyses are sound, and the experiments are well controlled. The results are presented with appropriate metrics (precision, recall, F1 score), which are standard in this field.

3. Conclusions are well stated, linked to original research question & limited to supporting results.
The conclusions are well stated and directly linked to the research question. They are appropriately limited to the supporting results, and the manuscript does not overstate the findings.

Additional comments

The manuscript presents a significant advancement in the field of Chinese NER with the introduction of the SLEBERT model. While the results are promising, the authors should consider addressing the language and clarity issues identified, particularly in the introduction and methods sections. Additionally, improving the quality of figures and providing more recent literature references would enhance the overall quality of the manuscript.

Reviewer 2 ·

Basic reporting

I appreciate the authors for addressing the previously raised comments. I am pleased to report that the authors have adequately addressed all the concerns and suggestions I raised in my previous review. The revisions have strengthened the manuscript, and I find the current version to be clear, comprehensive, and well-organized. I recommend the manuscript for acceptance in its current form.

Experimental design

No Comment.

Validity of the findings

No Comment.

Additional comments

No Comment.

Reviewer 3 ·

Basic reporting

NA

Experimental design

NA

Validity of the findings

NA

Additional comments

NA

---

## Round 0.3 · accepted · Accept

Thank you so much for your quick responses to the reviewers' comments. We have no further concerns or suggestions and the manuscript has been recommended for publication.